# Mechanical stretch scales centriole number to apical area via Piezo1 in multiciliated cells

**Saurabh Kulkarni[1†]\*, Jonathan Marquez[1], Priya Date[1‡], Rosa Ventrella[2], Brian J Mitchell[2], Mustafa K Khokha[1]\***

[1]Pediatric Genomics Discovery Program, Department of Pediatrics and Genetics, Yale University School of Medicine, New Haven, United States; [2]Department of Cell and Developmental Biology, Feinberg School of Medicine, Northwestern University, Chicago, United States

**Abstract** How cells count and regulate organelle number is a fundamental question in cell biology. For example, most cells restrict centrioles to two in number and assemble one cilium; however, multiciliated cells (MCCs) synthesize hundreds of centrioles to assemble multiple cilia. Aberration in centriole/cilia number impairs MCC function and can lead to pathological outcomes. Yet how MCCs control centriole number remains unknown. Using *Xenopus*, we demonstrate that centriole number scales with apical area over a remarkable 40-fold change in size. We find that tensile forces that shape the apical area also trigger centriole amplification based on both cell stretching experiments and disruption of embryonic elongation. Unexpectedly, Piezo1, a mechanosensitive ion channel, localizes near each centriole suggesting a potential role in centriole amplification. Indeed, depletion of Piezo1 affects centriole amplification and disrupts its correlation with the apical area in a tension-dependent manner. Thus, mechanical forces calibrate cilia/centriole number to the MCC apical area via Piezo1. Our results provide new perspectives to study organelle number control essential for optimal cell function.

**\*For correspondence:**
sk4xq@virginia.edu (SK);
Mustafa.khokha@yale.edu (MKK)

**Present address:** [†]Department of Cell Biology, Department of Biology, Center for Membrane and Cell Physiology, University of Virginia, Charlottesville, United States; [‡]College of Arts and Sciences, University of Virginia, Charlottesville, United States

## Introduction

Organelles compartmentalize cells into discrete functioning units. Cells must regulate the number of organelles to achieve proper function (*Marshall, 2007*; *Marshall, 2016*; *Nigg and Holland, 2018*; *Rafelski and Marshall, 2008*). For example, multiciliated cells (MCCs) line the epithelia of the brain ventricles, the airway, and the oviduct where motile cilia propel extracellular fluid to circulate cerebrospinal fluid, remove pathogens, and move the ova (*Spassky and Meunier, 2017*). Depending on the location, MCCs synthesize between 30 and 300 motile cilia (*Spassky and Meunier, 2017*). Assembly of too few or too many cilia impairs MCC function and is associated with several diseases including Primary Ciliary Dyskinesia, suggesting the existence of an active mechanism that controls cilia number (*Boon et al., 2014*; *Spassky and Meunier, 2017*; *Wallmeier et al., 2014*). Yet, the cellular and molecular mechanisms that control the number of cilia in MCCs remain unknown.

To shed light on mechanisms, we used the *Xenopus* embryonic epidermis, an established, versatile, in vivo model to study MCCs (*Walentek and Quigley, 2017*; *Werner and Mitchell, 2013*). There, MCCs are first specified in the basal epithelia, where they begin to synthesize centrioles using specialized structures called deuterosomes (*Figure 1a*, Step 0) (*Klos Dehring et al., 2013*; *Zhao et al., 2013*). As MCCs intercalate into the outer epithelial cell layer and expand their apical surface, centrioles migrate apically, dock at the apical surface, and provide the platform for assembly of motile cilia (*Figure 1a*, Steps 1–4) (*Deblandre et al., 1999*; *Kulkarni et al., 2018a*; *Stubbs et al., 2006*; *Zhang and Mitchell, 2015*). As such, in this study, we focused our efforts on counting

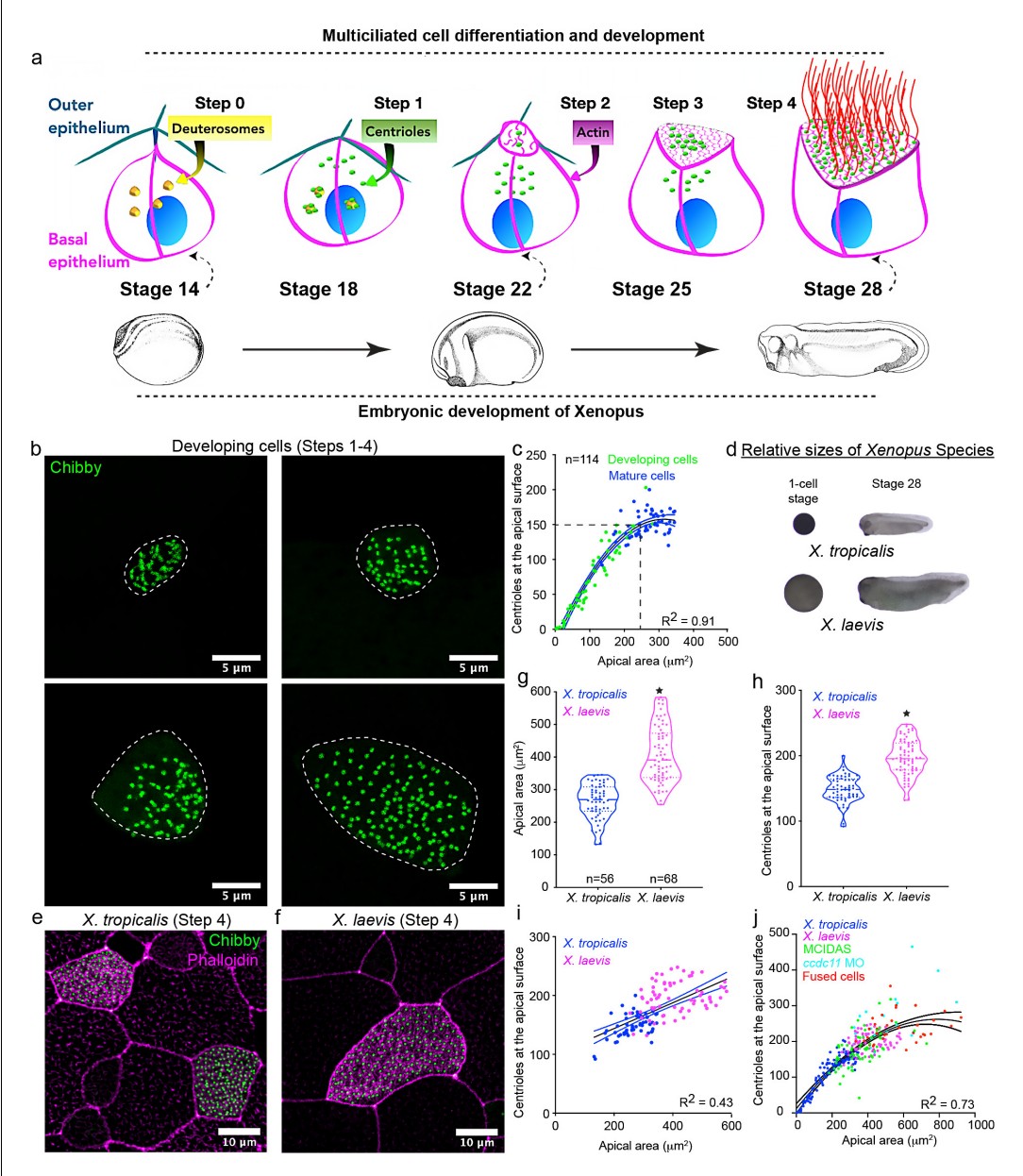

**Figure 1.** Centriole number scales with apical area in *Xenopus* MCCs. (**a**) Schematic representing the current understanding of how MCCs differentiate and develop (Steps 0–4). *Xenopus* embryonic development is closely linked (dashed arrows) to MCC development. (**b**) MCCs captured in different stages of development (Steps 2–4) and labeled with chibby-GFP (centrioles). Dotted line represents the cell boundary. (**c**) Regression plot showing the positive correlation between apical area and number of centrioles at the apical surface in developing (green, Step 2, 3) and mature MCCs (blue, Step 4). (**d**) One-cell stage and stage 28 embryos of *Xenopus tropicalis* and *Xenopus laevis*. Images are to scale. Mature (Step 4) epidermal MCCs marked with chibby-GFP (centrioles) and phalloidin (F-actin) of (**e**) *X. tropicalis* and (**f**) *X. laevis* embryos at stage 28. Quantitation of (**g**) apical area and (**h**) centriole number in MCCs of *X. tropicalis* and *X. laevis*. The statistical comparison between the treatments is done using an unpaired t test. (**i**) Regression plot showing the positive correlation between apical area and centriole number across species in mature MCCs. (**j**) Regression plot showing the scaling relationship exists over a 40-fold change in apical area among different treatments. $R^2$ is the correlation coefficient. * indicates statistical significance at $p < 0.05$. n = number of cells from 15 to 25 embryos. The data is uploaded as source data 1.

The online version of this article includes the following source data and figure supplement(s) for figure 1:

**Source data 1.** Source data related to *Figure 1* .

**Figure supplement 1.** Relation between centriole number and apical area in goblet cells converted to MCCs in *Xenopus laevis*.

**Figure supplement 2.** Increasing apical area leads to an increase in centriole number.

centrioles as a simple, efficient proxy for cilia number using *chibby-GFP*, a marker for mature centrioles (*Burke et al., 2014*).

## Results and discussion

MCC development is closely linked to embryonic development (*Figure 1a*). Therefore, we collected embryos at different developmental stages (from stage 20 to stage 28) to examine MCCs at various stages of apical expansion, ranging from MCCs that have just intercalated (Step 2) to fully mature MCCs (step 4). We measured the number of centrioles at the apical surface and the apical area. Surprisingly, we observed a strong correlation between the apical area and the number of centrioles at the apical surface (*Figure 1b,c*).

Next, we wanted to test if this relationship would persist if the MCC apical area became larger. We employed four different approaches to increase the apical area of MCCs. First, two species of *Xenopus* are common models for cell biology (*X. laevis* and *X. tropicalis*), and due to the evolutionary variation in embryonic sizes of the two species, they are useful for scaling experiments (*Figure 1d* – compare relative sizes of the eggs and embryo) (*Levy and Heald, 2012*). Compared to *X. tropicalis*, the *X. laevis* embryo is larger with significantly larger MCCs (*Figure 1d–g*, median ± SD, 391 ± 86 $\mu m^2$ vs. 270 ± 53 $\mu m^2$ in *X. tropicalis*). *X. laevis* MCCs also have significantly more centrioles (*Figure 1h*, median ± SD, 195 ± 27 compared to 150 ± 20 in *X. tropicalis*). Interestingly, by combining data from both species, we observed a clear trend where centriole number scales with MCC apical area (*Figure 1i*).

Second, in *X. laevis*, we converted epithelial goblet cells (which normally secrete mucus) to MCCs by overexpressing the master regulator of multiciliogenesis, *mcidas* (multiciliate differentiation and DNA synthesis-associated cell cycle protein) (*Figure 1—figure supplement 1a*; *Stubbs et al., 2012*). These induced MCCs in *X. laevis* are larger than *X. tropicalis* MCCs and have proportionately more centrioles (*Figure 1—figure supplement 1b,c*). Interestingly, the apical area of mature MCCs and the induced MCCs of *X. laevis* were similar and so were the number of centrioles (*Figure 1—figure supplement 1c*).

Third, we induced cytokinesis defects by knocking down *ccdc11* using a morpholino oligo (MO) in *X. tropicalis* (*Kulkarni et al., 2018b*). MCCs are mitotically mature; however, if their progenitor fails to undergo cytokinesis, then the resultant MCC can be much larger (*Figure 1—figure supplement 2a*). With this strategy, we identified MCCs with significantly larger apical areas (median ± SD, 437 ± 154 $\mu m^2$ vs. 202 ± 38 $\mu m^2$ in controls) (*Figure 1—figure supplement 2a,b*), and these cells had proportionately more centrioles (median ± SD, 227 ± 74 vs. 138 ± 26 in controls) (*Figure 1—figure supplement 2c,d*).

Finally, in *X. laevis*, we fused MCCs with neighboring (most likely) non-MCCs (confirmed by the presence of two nuclei – dashed lines in *Figure 1—figure supplement 2e*), resulting in much larger cells with increased apical area (median ± SD, apical area of 534 ± 149 $\mu m^2$ vs. 337 ± 49 $\mu m^2$ in controls) and more centrioles (median ± SD, 235 ± 40 vs. 168 ± 21 in controls) (*Figure 1—figure supplement 2e–h*). Interestingly, in each experiment, centriole number increased in proportion to the apical area suggesting that centriole amplification is a plastic process and cells can calibrate centriole number in response to cell size perturbations. By combining the data from controls and manipulated embryos, we found that this scaling relationship could be observed over a 40-fold change in the apical area, with the smallest apical area being ~25 $\mu m^2$ and the largest about 1000 $\mu m^2$ (*Figure 1j*). However, these experiments are limited in two ways. First, in our experiments, we increased the entire volume of the cell not just the apical area. Additionally, in cells with a cytokinesis defect or cell-cell fusion, we have combined two or more cells leading to an increase in the centriole number. Nevertheless, despite these limitations, the correlation between apical area and centriole number appears robust, and in subsequent experiments, we strived to overcome these limitations.

While these experiments suggest that the apical area of an MCC may regulate centriole number, we sought to test the alternative hypothesis that centriole number may determine apical area or that there may be a feedback mechanism between centriole number and apical area. We can manipulate centriole number in two ways: increase the number of centrioles by overexpressing *cep152* (*Collins et al., 2020*; *Klos Dehring et al., 2013*) or decrease the number of centrioles with Centrinone treatment, a PLK4 inhibitor (*Wong et al., 2015*). We first increased the centriole number by overexpressing *cep152* in *X. laevis* (*Figure 2a*), which increased the number of centrioles (median ±

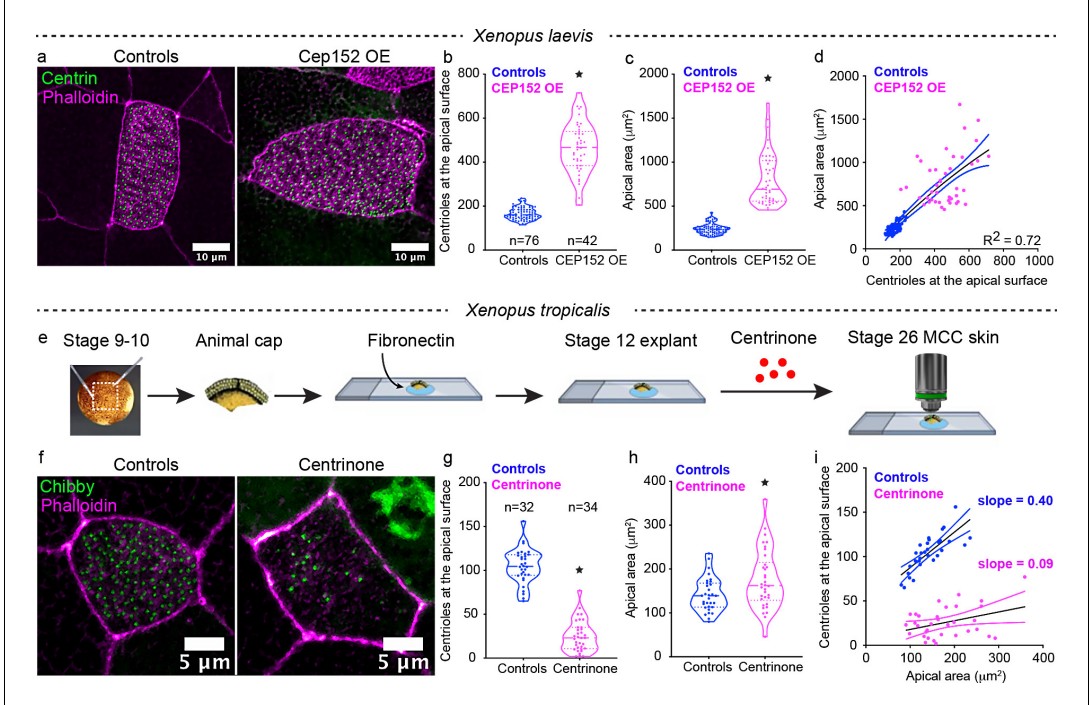

**Figure 2.** Perturbation of centriole amplification affects apical area contextually. (**a**) Mature (Step 4) epidermal MCCs marked with chibby-GFP (centrioles, green), and phalloidin (F-actin, magenta) in control and *Cep152* overexpressed (OE) embryos. Quantitation of (**b**) apical area and (**c**) centriole number in MCCs of control and CEP 152 OE embryos at stage 28. (**d**) Regression plot showing the positive correlation between apical area and centriole number. (**e**) Experimental design to block centriole amplification in MCCs using Centrinone in animal caps. We dissected the animal caps at stage 8–9 and tethered them to slides using fibronectin. At stage 14 (based on unmanipulated sibling embryos), we exposed the caps to Centrinone until their unmanipulated sibling embryos reached stage 25–26. (**F**) Epidermal MCCs marked with chibby-GFP (centrioles, green), and phalloidin (F-actin, magenta) in control and Centrinone-treated animal caps. Quantitation of (**g**) apical area and (**h**) centriole number in MCCs of control and Centrinone-treated animal caps. (**i**) Regression plot showing the loss of correlation between apical area and centriole number in Centrinone-treated MCCs. * indicates statistical significance at p < 0.05. The statistical comparison between the treatments (**b, c, g, h**) is done using an unpaired t test. $R^2$ is the correlation coefficient. n = number of cells collected from 10 to 15 embryos. The data is uploaded as source data 2.

The online version of this article includes the following source data for figure 2:

**Source data 1.** Source data related to *Figure 2*.

SD, 467 ± 113 vs. 160 ± 27 in controls) and was accompanied by a correlated increase in the apical area (median ± SD, 692 ± 296 μm$^2$ vs. 236 ± 58 μm$^2$ in controls) (*Figure 2a–d*). Next, we reduced centriole numbers with Centrinone. Centriole synthesis begins during intercalation, when the cells are in the basal layer (*Figure 1a*, Step 0). To allow the chemical inhibitor to access the cells in the basal layer, we generated *Xenopus* embryonic 'stem cell' explants (commonly referred to as animal caps), which auto-differentiate into an embryonic epidermis replete with MCCs (*Figure 2e*). We harvested animal caps from *X. tropicalis* embryos and grew them on fibronectin-coated slides with exposure to Centrinone or vehicle alone until control embryos reached stage 25–26 (*Figure 2e,f*). We successfully reduced the median number of centrioles from 104 in controls to 25 in Centrinone-treated MCCs (*Figure 2f,g*). Despite a dramatic reduction in the number of centrioles, we found a slight *increase* in the apical area of Centrinone-treated MCCs as compared to controls (median ± SD, 175 ± 67 vs. 141 ± 38 μm$^2$ in controls) (*Figure 2h*). From this result, we conclude that a minimum apical size can be achieved independent of the centriole amplification (*Figure 2f–i*). Once this minimum size is reached, then centrioles may contribute to apical expansion (*Figure 2a–d*). Moving forward, we focused on the hypothesis that the apical area may fine tune centriole number.

While we initially focused on the number of centrioles at the apical surface, this may not reflect the total number of centrioles in the cell. Previous studies have noted the presence of centrioles in the intercalating MCCs (*Figure 1a*, Steps 0–1), but the number of these centrioles is unknown (*Collins et al., 2020*; *Werner et al., 2014*). One possibility is that MCCs assemble all of the

centrioles (~150 in *X. tropicalis*) during intercalation with subsequent waves of either synthesis or degradation depending on the final apical area. Alternatively, MCCs may continuously produce centrioles and halt this process based on the final apical area. In either model, MCCs would require a cellular and molecular mechanism to measure the apical area.

To differentiate between these hypotheses, we set out to image centrioles in the intercalating cells. To image deep within the cytoplasm, we used animal caps which are relatively transparent compared to whole embryos and imaged centrioles (Chibby-GFP) along the apico-basal axis of intercalating MCCs. Using segmentation and 3D reconstruction, we could observe the process of centriole migration and count all the centrioles in the cytoplasm of intercalating MCCs (presumptive MCC border marked by white dotted line, *Figure 3a–d*, *Videos 1* and *2*). In these intercalating cells, the number of centrioles was 75 (median), approximately half the number in mature *X. tropicalis* MCCs (*Figure 3e*, blue, Step 4; magenta, Steps 0–1). Interestingly, *X. laevis* MCCs also make 90 (median) centrioles during intercalation, again half the number of centrioles in mature MCCs (*Figure 3f* blue, Step 4; magenta, Steps 0–1). From these results, we conclude that (1) *Xenopus* MCCs synthesize half the total number of centrioles prior to intercalation, (2) these centrioles dock to the apical surface in a manner that scales with apical area, and (3) the remaining half of the number of centrioles must be synthesized in a second round that is regulated based on the apical area.

The apical area of the cell is dependent on multiple parameters including but not limited to the overall size of the cell, cell autonomous pushing forces, as well as pulling forces by neighboring cells (*Guillot and Lecuit, 2013*; *Heisenberg and Bellaïche, 2013*; *Mao and Baum, 2015*;

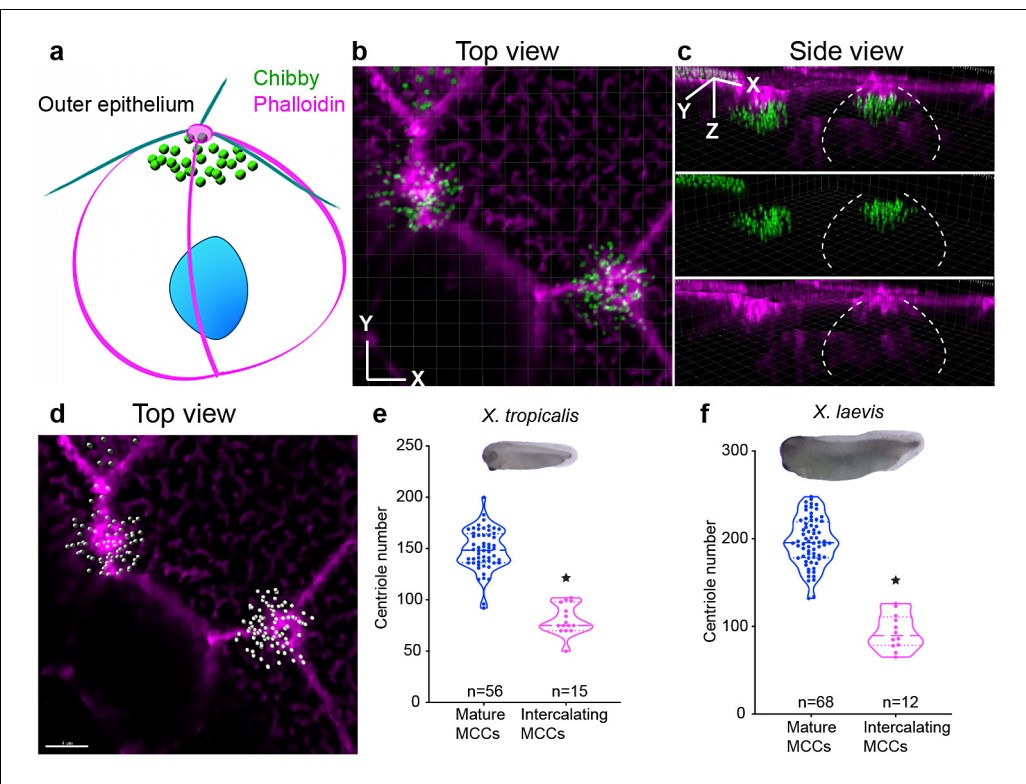

**Figure 3.** MCCs synthesize about half the total number of centrioles during intercalation. (**a–c**) Centrioles (chibby-GFP, green) and F-actin (phalloidin, magenta) in intercalating MCCs (Step 1). Dotted while lines show the border of the intercalating MCCs. (**d**) The same MCC is segmented using IMARIS to show individual centrioles in grey and F-actin in magenta. MCCs generate half of the total number of centrioles just prior to intercalation in (**e**) *X. tropicalis* and (**f**) *X. laevis*. * indicates statistical significance at p < 0.05. The statistical comparison between the treatments is done using an unpaired t test. n = number of cells from 10 to 15 embryos/species. The data is uploaded as source data 3.

The online version of this article includes the following source data for figure 3:

**Source data 1.** Source data related to *Figure 3*.

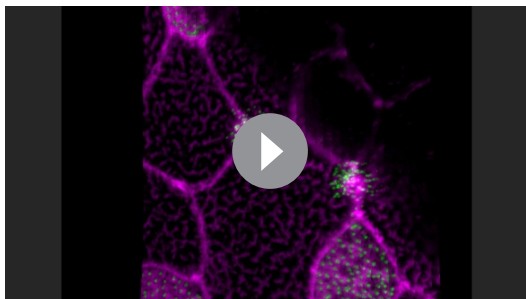

**Video 1.** Centrioles (green) and F-actin (magenta). Centrioles dispersed below the apical surface of an intercalating MCC.

https://elifesciences.org/articles/66076#video1

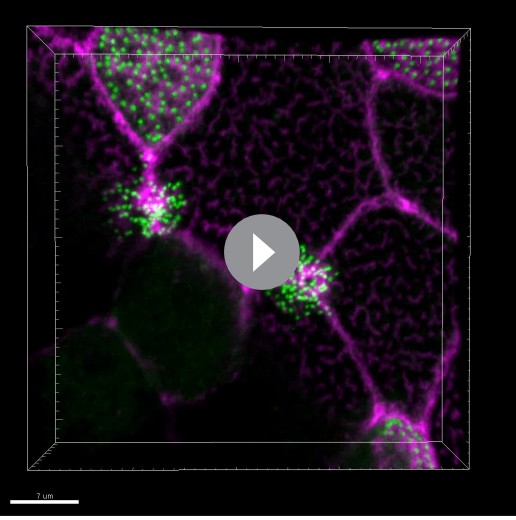

**Video 2.** Segmentation and 3D reconstruction of the *Video 1* using IMARIS.

https://elifesciences.org/articles/66076#video2

*Sedzinski et al., 2016*). Specifically, during intercalation, the MCC is thought to cell autonomously push against its neighbors to expand its apical surface. Subsequently, neighboring cells pull on the MCC at cell junctions, expanding the apical area further (*Sedzinski et al., 2016*). To elucidate the contributions of pushing vs. pulling forces, we decided to examine the shape of the cells during apical expansion. Cell autonomous pushing forces would be radially symmetric so the apical surface should expand circularly (*Sedzinski et al., 2016*). On the other hand, cell non-autonomous pulling forces would depend on the relative positions of the neighboring cells and cell junctions leading to a polygonal apical shape (*Sedzinski et al., 2016*). The thinness ratio (TR) which relates the area of a shape to the square of its perimeter can detect these changes in cell shapes (*Figure 4a*). The TR is 1 for a circle and < 1 for polygons (*Figure 4a*). When we plotted the TR as a function of apical area, we found that MCCs with small apical areas have a TR of nearly 1, while the TR decreases to 0.8 as the apical area increases to ~300 µm² (*Figure 4b,c*, *Video 3*). Therefore, TR measurements support the notion that the initial apical expansion is driven by cell autonomous pushing forces, while subsequent apical expansion is driven largely by cell non-autonomous pulling forces.

We speculated that cell non-autonomous pulling forces that drive the final phase of apical expansion might define the apical area and centriole number in MCCs. To test the hypothesis, we began with an embryological approach to manipulate the apical area. In a developing embryo, morphogenetic movements create forces that lead to dramatic shape changes that transform a spherical embryo (stage 9–10) to an elongated one (stage 28), presumably, exerting stretching forces on the epidermal MCCs to increase their apical area (*Figure 1a*). For example, Spemann's Organizer, which is dependent on Wnt signaling, dorsalizes the mesoderm and ectoderm, which subsequently creates considerable elongation forces (*De Robertis et al., 2000*; *Harland and Gerhart, 1997*; *Hikasa and Sokol, 2013*; *Keller and Sutherland, 2020*; *Kiecker, 2000*). By depleting β-catenin, a key effector of the Wnt signaling pathway, we can eliminate the formation of Spemann's Organizer and generate cylindrically symmetric embryos that lack dorsal structures and have much less elongation compared to control embryos (*Figure 4—figure supplement 1a,c*, *Heasman et al., 1994*; *Khokha et al., 2005*). While β-catenin-depleted embryos can form functioning MCCs that generate fluid flow, both the MCCs (median ± SD, 106 ± 26 µm² vs. 267 ± 64 µm² in controls) and non-MCCs (median ± SD, 319 ± 104 µm² vs. 428 ± 125 µm² in controls,) have smaller apical areas (*Figure 4—figure supplement 1b,d–f*). Interestingly, the centriole number in these embryos is also significantly decreased (median ± SD, 100 ± 16 vs. 148 ± 26 in controls, *Figure 4—figure supplement 1g,h*), approaching the 75 centrioles formed prior to intercalation. Further, the TR in β-catenin-depleted MCCs is significantly higher and closer to 1 (median ± SD, 0.93 ± 0.05 vs. 0.77 ± 0.04 in controls, *Figure 4—figure supplement 1i,j*), supportive of a significant reduction of pulling forces exerted on MCCs. This result suggests that the lack of embryonic elongation forces in β-catenin-depleted embryos causes the reduction in the MCC apical area and centriole number. However, a challenge in this experiment is

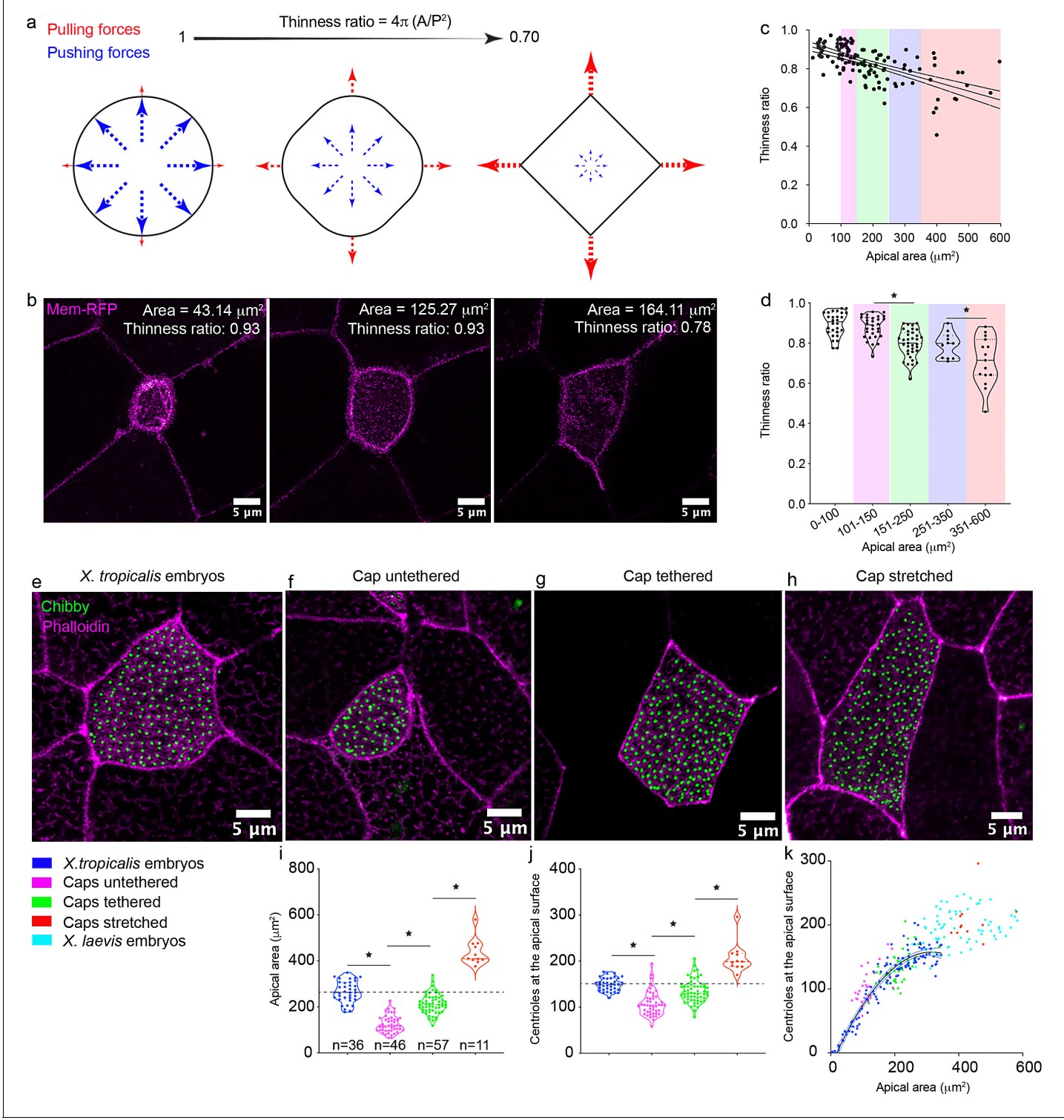

**Figure 4.** Mechanical stretch triggers centriole amplification in MCCs. (**a**) Schematic showing the effect of cell autonomous pushing (blue) vs. cell non-autonomous pulling forces (red) on cell shape and the thinness ratio (TR). (**b**) A single MCC (marked by membrane-RFP) undergoing apical expansion and in the process changing the cell shape from circular (TR=0.93) to more polygonal (TR=0.78). (**c**) A regression plot showing the negative correlation between the TR and the apical area. Magenta: 100–150 $\mu m^2$, Green: 151–250 $\mu m^2$, Blue: 251–350 $\mu m^2$, Red: 351–600 $\mu m^2$. n = 122 cells collected from 20 to 25 embryos (**d**) Binning the apical area shows that the increase in apical area leads to a significant reduction in the TR. MCCs marked with chibby-GFP (centrioles, green), and phalloidin (F-actin, magenta) in (**e**) control embryos, (**f**) untethered animal caps, (**g**) tethered animal caps, and (**h**) mechanically stretched animal caps. The statistical comparison between the treatments is done one-way ANOVA test followed by Tukey's multiple

*Figure 4 continued on next page*

*Figure 4 continued*

comparisons test. Quantitation of (**i**) apical area and (**j**) centriole number in MCCs of animal caps subjected to different mechanical stimuli. Dashed line indicates the median value of controls. * indicates statistical significance at p < 0.05. The statistical comparison between the treatments is done using the Brown-Forsythe and Welch ANOVA test followed by the Dunnett's T3 multiple comparisons test. n = number of cells. Data for untethered and tethered caps was collected from 10 to 12 animal caps. Data for stretched animal caps was collected from six to nine animal caps. (**k**) Regression plot demonstrates the scaling relation between the apical area and centriole number across different treatments. The data is uploaded as source data 4.
The online version of this article includes the following source data and figure supplement(s) for figure 4:

**Source data 1.** Source data related to *Figure 4*.
**Figure supplement 1.** Depletion of β-catenin affects apical area and centriole number in MCCs Stage 28 embryos of (**a**) controls and (**c**) *β-catenin* morphants.
**Figure supplement 2.** Manipulation of animal caps by cell adhesion and mechanical stretcher.
**Figure supplement 3.** Apical area of non-MCCs in untethered/tethered animal caps.

the confounding effects generated by genetic manipulations, such as diminished Wnt signaling or potential changes in cell adhesion in β-catenin depleted embryos.

To avoid these confounding effects, we sought to manipulate MCCs using non-genetic tools. We returned to animal caps, *Xenopus* stem cell explants that auto-differentiate into an embryonic multi-ciliated epidermis and raised them in two different conditions. In the first condition, we harvested animal caps and cultured them on agarose. In this case, because the cells do not adhere to agarose, the animal caps roll up to form irregular spherical structures which we called 'untethered' explants (*Figure 4—figure supplement 2a,b*, *Video 4*). The MCCs in these explants had an apical area just slightly larger than in β-catenin-depleted embryos (median ± SD, 116 ± 37 μm² compared to 106 ± 26 μm² in β-catenin-depleted embryos and 263 ± 46 μm² in controls, *Figure 4e,f,i*) and a correlated decrease in centriole number (median ± SD, 105 ± 27 compared to 100 ± 16 in β-catenin-depleted embryos and 149 ± 14 in controls, *Figure 4j*). In the second condition, we harvested animal caps and cultured them on fibronectin-coated slides. In this case, the cells adhere to the slide and spread outward (*Stepien et al., 2019*). As a result, these 'tethered' explants are stretched along the slide to form flat epithelia (*Figure 4—figure supplement 2c,d*, *Video 5*; *Stepien et al., 2019*). In these tethered explants, the apical area of both non-MCCs (*Figure 4—figure supplement 3*, median ± SD, 372 ± 159 μm² compared to 170 ± 86 μm² in untethered caps and 465 ± 177 μm² in controls) and the MCCs (*Figure 4e–g,i*, median ± SD, 210 ± 43 μm² compared to 116 ± 37 μm² in untethered caps compared to 263 ± 46 μm² in controls) are increased compared to the untethered caps but are slightly smaller than epithelial cells in the embryo suggesting that additional forces or factors in the embryo may contribute to the apical area. Nevertheless, the tethered caps had a significant increase in the number of centrioles (*Figure 4j*, median ± SD, 130 ± 25 vs 105 ± 27 in untethered caps compared to 149 ± 14 in controls) in an area-dependent manner.

To understand the contribution of pulling forces in defining the apical area, we analyzed cell shapes and measured the TR. Specifically, by binning the data based on our results, from 0 to 100 μm² (apical areas of MCCs in the initial stages of development, step 2, median ± SD, TR: 0.90 ± 0.05), 100–150 μm² (untethered caps, median ± SD, TR: 0.89 ± 0.05), 150–250 μm² (tethered caps, median ± SD, TR: 0.79 ± 0.06), 250–350 μm² (wildtype *X. tropicalis* MCCs, median ± SD, TR: 0.79 ± 0.06), the TR reduces significantly as the cells become larger, supporting the increasing contribution of pulling forces on defining the apical area (*Figure 4d*). Taken together, these results suggested that tension generated by stretching within the epithelial

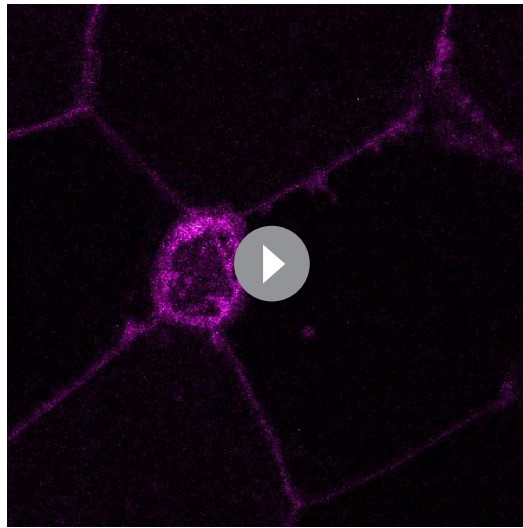

**Video 3.** MCC labeled with membrane-RFP undergoing expansion of its apical surface.
https://elifesciences.org/articles/66076#video3

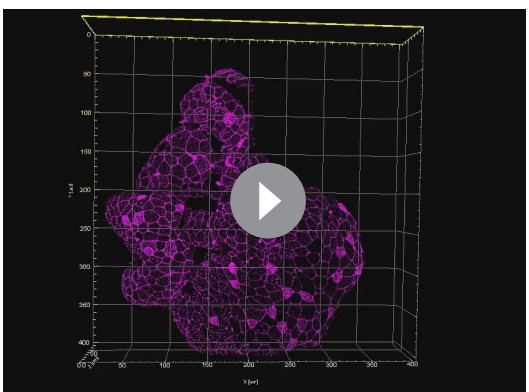

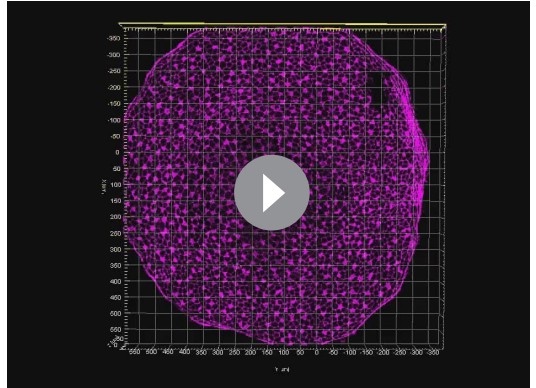

**Video 4.** Untethered cap forms an irregular spherical structure. F-actin is in magenta.
https://elifesciences.org/articles/66076#video4

**Video 5.** Tethered animal cap forms a flat multiciliated epithelium. F-actin is in magenta.
https://elifesciences.org/articles/66076#video5

sheet is critical to achieve proper apical area and triggers centriole amplification over the initial set of 75 centrioles.

To directly test the role of stretching force on centriole number, we applied an artificial radial stretch to the explants. Specifically, we raised *X. tropicalis* explants on a silicone membrane coated with fibronectin until sibling control embryos reached stage 26. At this stage, MCCs are nearly mature, and we stretched the explants radially for 3 hr in a stepwise fashion (*Figure 4—figure supplement 2e,f*). This stepwise stretch created a force of 11.5 N and about 50–75% strain. We observed a significant increase in the apical area of MCCs (median ± SD, 409 ± 57 μm² compared to 210 ± 43 μm² in unstretched tethered caps) (*Figure 4g–i*). Stretching also led to a dramatic change in cell shape and a further significant reduction in the TR (apical area: 351–600 μm², median ± SD, TR: 0.71 ± 0.1, *Figure 4c,d*) compared to both tethered unstretched caps (median ± SD, TR: 0.79 ± 0.6) and WT *X. tropicalis* MCCs (median ± SD, TR: 0.79 ± 0.6), consistent with the expectation that external stretching will lead to more polygonality of apical shape. In these stretched MCCs, the number of centrioles also increased (median ± SD, 199 ± 33 vs 130 ± 25 in unstretched tethered caps) demonstrating that stretching forces trigger centriole amplification in an area dependent manner in MCCs (*Figure 4j*). Interestingly, just by stretching, we transformed MCCs in *X. tropicalis* explants to sizes more similar to *X. laevis* (median ± SD, 409 ± 57 μm² vs. 390 ± 87 μm² in *X. laevis*) and the number of centrioles generated were also similar (median ± SD, 199 ± 33 vs. 195 ± 27 in *X. laevis*), highlighting the conserved role of mechanical forces in establishing the scaling mechanisms across species (*Figure 1g–j*, *Figure 4k*).

Given the central role stretching plays in regulating centriole number, we decided to investigate the molecular mechanisms that sense the force. While there are several molecules that can act as mechanosensors (*Luo et al., 2013*; *Martino et al., 2018*; *Wang, 2017*), we were particularly struck by the punctate distribution pattern of Piezo1 (*Figure 5—figure supplement 1a*). Piezo1 is a mechanosensitive cation channel that responds directly to membrane stretch and is primarily expressed in epithelial cells exposed to fluid pressure and flow (*Bagriantsev et al., 2014*; *Wang and Xiao, 2018*; *Wu et al., 2017*). In addition to its expression at cell junctions (*Figure 5—figure supplement 1b* – dashed box), we unexpectedly discovered that Piezo1 is localized adjacent to the centrioles at the apical membrane (*Figure 5a*, *Figure 5—figure supplement 1a,b*). Piezo1 localization is diminished with MO-based Piezo1 depletion indicating that this anti-Piezo1 antibody signal is specific (*Figure 5—figure supplement 1c–e*).

To test the role of Piezo1 in regulating centriole number in MCCs, we depleted Piezo1 using MO and CRISPR and also inhibited its activity using GSMTx4, a spider venom peptide that inhibits cationic mechanosensitive channels including Piezo1 (*Gnanasambandam et al., 2017*). Centrioles were measured in mature MCCs and all three treatments resulted in a significant decrease in centriole number compared to control embryos (*Figure 5b,d–g*, median ± SD, 105 ± 22 *piezo1* MO, 116 ± 24 *piezo1* CRISPR, and 110 ± 26 GSMTx4, compared to 140 ± 21 in uninjected controls and 133 ± 22 in standard MO controls). This reduction in centriole number did not appear to be due to a defect in

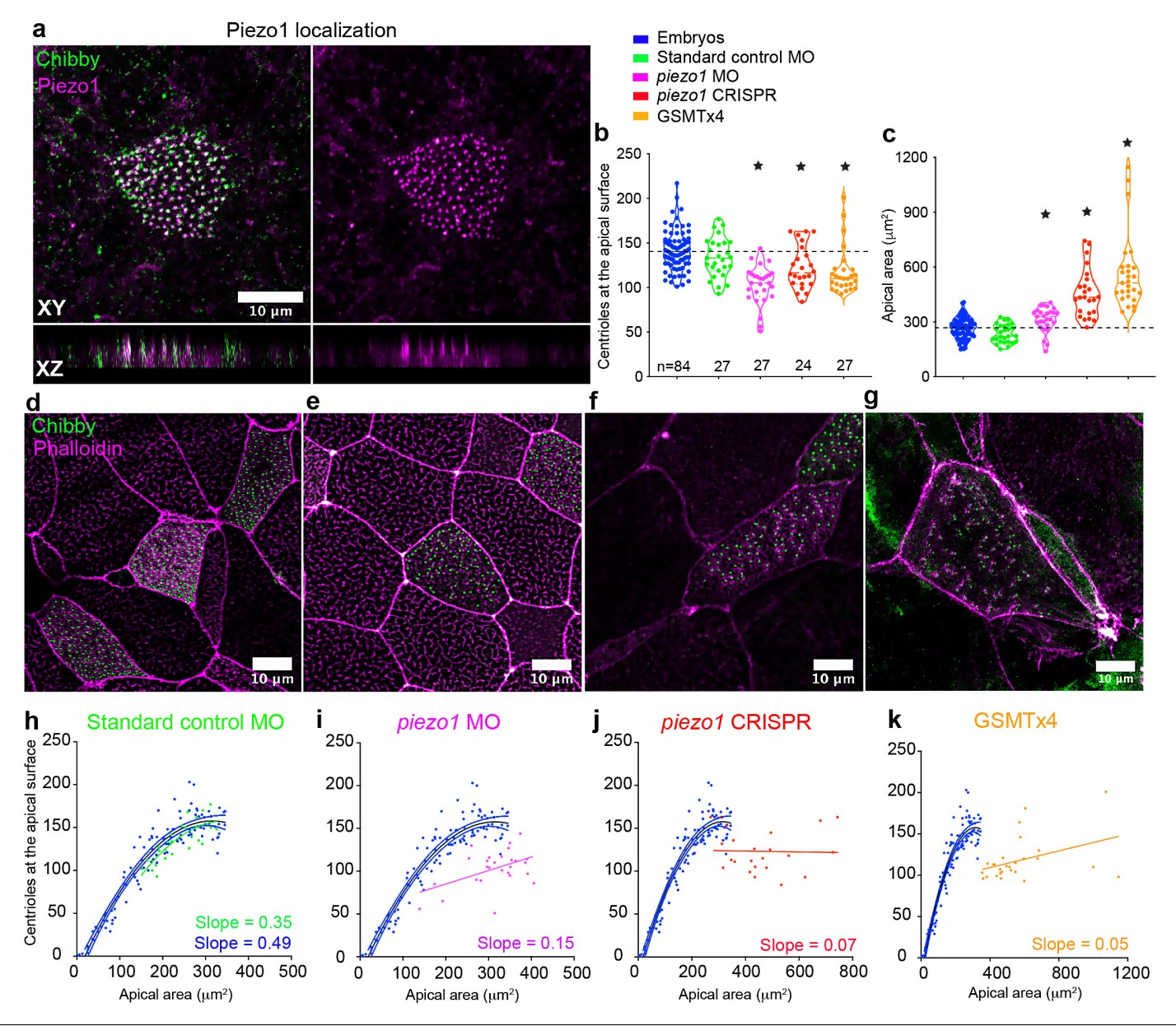

**Figure 5.** Piezo1 fine tunes centriole amplification and the scaling relation with apical area in the embryos. (a) Mature epidermal MCCs marked with anti-Piezo1 antibody (magenta) and chibby-GFP (centrioles, green) in *X. tropicalis* embryos. XZ axis shows that Piezo1 localizes at the same plane as centrioles. Quantitation of (b) centriole number and (c) apical area in MCCs across different treatments that affect Piezo1 levels (MO or CRISPR) or function (GSMTx4). Dashed lines indicate the median values of controls. * indicates statistical significance at $p < 0.05$. The statistical comparison between the treatments is done using the Brown-Forsythe and Welch ANOVA test followed by Dunnett's T3 multiple comparisons test. n = number of cells collected from 12 to 20 embryos. MCCs marked with chibby-GFP (centrioles), and phalloidin (F-actin) in (d) Standard control MO, (e) *piezo1* MO, (f) *piezo1* CRISPR, and (g) GSMTx4. Regression plot demonstrating the positive correlation between apical area and centriole number in mature MCCs of controls (blue) and (h) standard control MO (green) compared to loss of correlation in (i) *piezo1* MO (magenta), (j) *piezo1* CRISPR (red), and (k) GSMTx4 (orange). The data is uploaded as source data 5.

The online version of this article includes the following source data and figure supplement(s) for figure 5:

**Source data 1.** Source data related to *Figure 5*.

**Figure supplement 1.** Piezo1 localization at the cell junctions and the bases of cilia in the MCCs.

docking as we did not detect centrioles inside the cell. Further, Piezo1 depletion also uncoupled the relationship between apical area and centriole number as evident by the *increase* instead of a decrease in the apical area (*Figure 5c–g*) and flattening of the regression line in the treated embryos (*Figure 5h–k*). Our results demonstrate that Piezo1 is essential for calibrating centriole number in relation to the apical area in MCCs.

In Piezo1-depleted embryos, the centriole number in MCCs was similar to β-catenin depleted MCCs (median ± SD 100 ± 16) and untethered animal caps (median 105 ± 27), both of which experienced diminished pulling forces. Thus, our data suggest that Piezo1 is not essential to generate the first 100 centrioles but calibrates the final 50 centrioles in response to pulling forces. To test this hypothesis, we depleted Piezo1 and raised animal caps in untethered and tethered conditions. In the context of untethered animal caps, centriole number did not significantly differ in controls and Piezo1 depleted MCCs (*Figure 6a,c*, median ± SD, 101 ± 15 vs. 94 ± 15). In contrast, in tethered animal caps, Piezo one depletion led to a significant reduction in centriole number compared to controls (*Figure 6b,d*, median ± SD, 119 ± 14 in controls vs. 109 ± 16). Interestingly, with Piezo1 depletion, the number of centrioles in MCCs of tethered caps was similar to wild-type MCCs of untethered caps (median 109 in Piezo1 depleted compared to 101 in controls), demonstrating that Piezo1 is required for stretch-induced centriole amplification in the MCCs of *Xenopus*. Taken together, our data demonstrated that the stretching of MCCs due to morphogenetic movements calibrates centriole number in proportion to the apical area via Piezo1.

MCCs must regulate the number of cilia to optimize extracellular fluid flow. While a previous study using mouse tracheal epithelial cell culture suggested a correlation between cilia number and the apical area (*Nanjundappa et al., 2019*), the underlying mechanism was unknown. Our study, using the *Xenopus* embryonic epidermis, demonstrates that MCC apical area undergoes dramatic size changes as cell non-autonomous forces generated by morphogenetic movements pull on the epithelia. Importantly, centriole amplification occurs while the MCCs are being stretched and expanding their apical surface. Therefore, the cells must contend with a complex mathematical problem: how to count centrioles, how to measure the apical area, and how to coordinate the two. Our results show that Piezo1 translates the pulling forces that define the apical area into an appropriate number of centrioles (*Figure 6f*). Thus, Piezo1-mediated mechanosensation couples apical area and centriole number (*Figure 6e,f*).

There are a few possibilities that may explain how Piezo1 calibrates centriole number and controls the correlation between centriole number and the apical area. One possibility is that stretch may activate Piezo1, which leads to an influx of $Ca^{2+}$ from the extracellular environment. This increase in intracellular $Ca^{2+}$ may promote centriole amplification via either transcriptional or non-transcriptional means. Alternatively, Piezo1 has been shown to regulate the expression of focal adhesion kinases (FAK, Paxillin and Vinculin) in cancer cells leading to changes in tissue stiffness (*Chen et al., 2018*). These three focal adhesion kinases localize to the bases of cilia in MCCs, and their downregulation causes defects in actin and cilia assembly (*Antoniades et al., 2014*). However, their role in the regulation of centriole number and apical area remains unexplored. Finally, filamentous (F)-actin plays a critical role in mechanotransduction in all cells (*Massou et al., 2020*; *Wang, 2017*). At the apical membrane, MCCs are enriched in F-actin and the apparent loss of apical F-actin in Piezo1-depleted cells may lead to defective mechanotransduction resulting in the defects in centriole amplification and apical area.

In our work, we exploited the frog multiciliated embryonic epithelium because of two main considerations. First, studying the effects of tissue scale forces on MCC apical area and cilia number requires an in vivo system and therefore is challenging in mammals. Second, we could exploit the two-step process of MCC formation (*Deblandre et al., 1999*; *Stubbs et al., 2006*), radial intercalation followed by apical expansion to determine the number of centrioles at time zero (just prior to apical expansion) (Step 1, *Figure 1a*, *Figure 2*). Then we could compare that number to the final count to understand the contribution of stretching forces on centriole amplification. While mammalian MCCs do not radially intercalate like *Xenopus* MCCs, they still scale centriole number to the apical area suggesting a conserved mechanism (*Nanjundappa et al., 2019*). Indeed, the formation of MCCs in the mammalian respiratory epithelium is similar to the goblet cells converted to MCCs by *mcidas* overexpression in *Xenopus*, which also scale centriole number to the apical area. Our results define a molecular pathway in which tissue scale forces regulate apical area and cilia number in

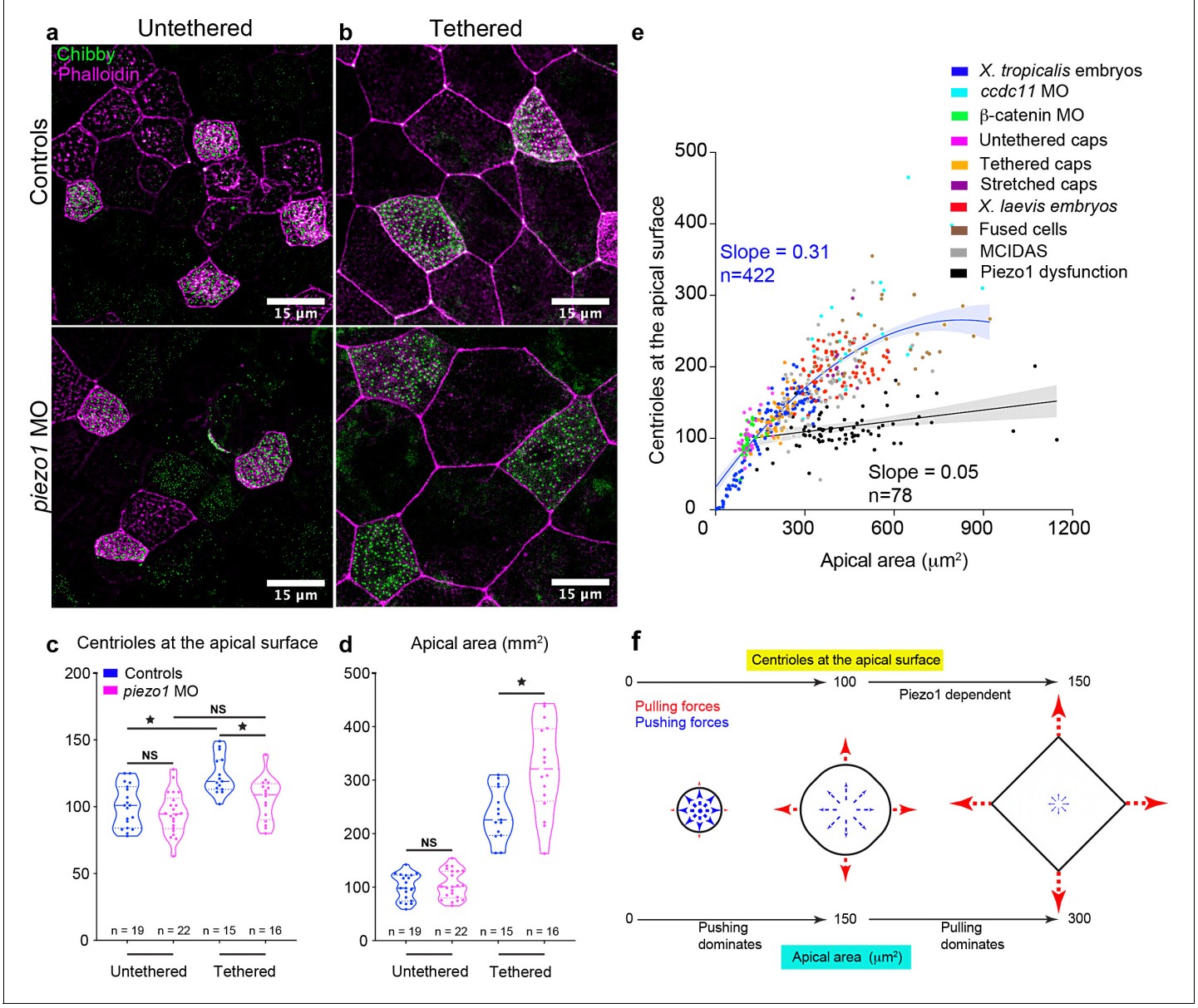

**Figure 6.** Piezo1 dysfunction leads to reduced number of centrioles in MCCs in a tension-dependent manner. MCCs marked with chibby-GFP (centrioles, green), and phalloidin (F-actin, magenta) in controls and *piezo1* morphants in (**a**) untethered animal caps and (**b**) tethered animal caps. Quantitation of (**c**) number of centrioles at the apical surface and (**d**) apical area in untethered animal caps and tethered animal caps. * indicates statistical significance at p < 0.05. The statistical comparison between the treatments is done using a one-way ANOVA followed by Tukey's multiple comparison test. n = number of cells collected from six to eight animal caps obtained from two independent experiments. (**e**) Regression plot showing a positive scaling relationship between apical area and the number of centrioles at the apical surface across all the treatments performed in this paper. When Piezo1 function is affected (black: *piezo1* MO, *piezo1* CRISPR, and GSMTx-4), the scaling relationship is abolished. n = number of cells. (**f**) Model illustrating that pushing forces dominate in the initial phase of apical expansion (apical area increases from 0 to ~150 µm²) and the centriole number reaches ~100. In the later phase of apical expansion, pulling forces dominate which changes the cell shape from round to polygonal. The pulling force is sensed by Piezo1 to regulate the amplification of the next ~50 centrioles. The data is uploaded as source data 6.

The online version of this article includes the following source data for figure 6:

**Source data 1.** Source data related to *Figure 6*.

MCCs. These results provide a new perspective to study the role of cell and tissue level mechanical forces in shaping organelle number to optimize cell function.

# Materials and methods

## Key resources table

| Reagent type (species) or resource | Designation | Source or reference | Identifiers | Additional information |
|---|---|---|---|---|
| Other | Translation blocking morpholino (*X. tropicalis*) | *b-catenin* | 5'-TTTCAACAGTTTCCAAAGAACCAGG-3' | 7.5–10 ng/ embryo |
| Other | Translation blocking morpholino (*X. tropicalis*) | *ccdc11* | 5'-CATGCTTTCTCCCCAGCCGTGCTGT-3' | 7.5–10 ng/ embryo |
| Other | Translation blocking morpholino (*X. tropicalis*) | *piezo1* | 5'- CACAGAGGACTTGCAGTTCCATC-3' | 10 ng/embryo |
| Other | Translation blocking morpholino (*X. tropicalis*) | standard control | 5'- CCTCTTACCTCAGTTACAATTTATA −3' | 10 ng/embryo |
| Other | CRISPR (*X. tropicalis*) | *piezo1* | 5'- GGGGCAGAAGGAGCCAAAAC −3' | 600 ng of sgRNA and 2.4 ng of NLS-Cas9 protein (PNABio)/ embryo |
| Antibody | Anti-Piezo1 (Rabbit polyclonal) | Novus | NBP1-78537 | IF (1:25) |
| Recombinant DNA reagent | Chibby-GFP (plasmid) | *Kulkarni et al., 2018b* | | |
| Recombinant DNA reagent | GFP-Centrin4 | *Klos Dehring et al., 2013* | | |
| Recombinant DNA reagent | RFP-Cep152 | *Klos Dehring et al., 2013* | | |
| Recombinant DNA reagent | hGR-Mcidas | *Stubbs et al., 2012* | | |
| Recombinant DNA reagent | GFP-Sas6 | *Stubbs et al., 2012* | | |
| Chemical compound, drug | GSMTx-4 | Abcam | ab141871 | |
| Chemical compound, drug | Centrinone | Tocris | 5687 | |
| Other | Alexa 488 Phalloidin | Thermo Fisher | A12379 | |
| Other | Alexa 647 Phalloidin | Thermo Fisher | A30107 | |

## Xenopus tropicalis

*Xenopus tropicalis* were housed and cared for in our aquatics facility according to established protocols approved by the Yale Institutional Animal Care and Use Committee (IACUC) and University of Virginia IACUC. Embryos were produced by in vitro fertilization. First, we harvested the testes of an adult male in 1x MBS + 0.2%BSA. Testes were then crushed and incubated with eggs for 3 min and then flooded with 0.1x MBS (pH 7.8–8) for 10 min. Fertilized eggs were then dejellied using 3% Cysteine in 1/9x MR (pH 7.8–8) for 6 min. Embryos were then washed using 1/9x MR and used for microinjections (described below) or raised to appropriate stages in 1/9x MR + gentamycin according to established protocols (*del Viso and Khokha, 2012*; *Khokha et al., 2002*). *Xenopus* tadpoles were staged according to the staging table previously described (*Nieuwkoop and Faber, 1994*). The developmental stages of embryos used for experiments are reported throughout the text and figures as appropriate.

## Xenopus laevis

*Xenopus laevis* were housed and cared for according to established animal care protocol approved by Northwestern University IACUC.

## Microinjection of MOs, CRISPR/Cas9 and mRNA and chemical inhibitor exposure in *Xenopus*

Morpholino oligonucleotides, CRISPR/Cas9, or mRNA were injected using a fine glass needle and Picospritzer system into one-cell or two-cell embryos, as described previously (*Khokha et al., 2002*). The following constructs were injected. *β-catenin* translation blocking MO: (7.5–10 ng/ embryo, 5′-TTTCAACAGTTTCCAAAGAACCAGG-3′), *ccdc11* translation blocking MO: (7.5–10 ng/ embryo, 5′-CATGCTTTCTCCCCAGCCGTGCTGT-3′), *piezo1* translation blocking MO: (10 ng/embryo 5′- CACA-GAGGACTTGCAGTTCCATC-3′), and the standard control MO (10 ng/embryo 5′- CCTCTTACC TCAGTTACAATTTATA −3′) was injected as a negative control. For F0 CRISPR, we generated sgRNAs using the EnGen sgRNA synthesis kit (NEB) following the manufacturer's instructions after creating a template Piezo1 sgRNA with the target site sequence of (5′- GGGGCAGAAGGAGC-CAAAAC −3′) as previously described (*Bhattacharya et al., 2015*). We then injected 600 ng of sgRNA and 2.4 ng of NLS-Cas9 protein (PNABio) into one-cell stage embryos. For mRNA injections, we generated in vitro capped mRNA using the mMessage machine kit (Ambion) and followed the manufacturer's instructions. Full-length Chibby-GFP (100 pg) RNA was injected into one-cell embryos of *X. tropicalis* to label centrioles. For *Cep152* overexpression, embryos were injected at the two- to four-cell stage with mRNA encoding GFP-Centrin4 and RFP-Cep152 (*Klos Dehring et al., 2013*). Embryos were allowed to develop until stage 28–30 and fixed in 3% PFA, followed by staining with Phalloidin and DAPI. For *Mcidas* overexpression, embryos were injected at the two- to four-cell stage with mRNA encoding GFP-Centrin4 or GFP-Sas6 together with hGR-Mcidas (*Stubbs et al., 2012*). Embryos were allowed to develop until stage 10.5 and then treated with 20 µM Dexamethasone and allowed to develop until stage 28–30. Embryos were fixed in 3% PFA, followed by staining with Phalloidin. GSMTx-4 treatment: After removing the vitellin envelope, stage 14–15 embryos were exposed to 15 µM of GSMTx4 until they reached stage 28. At stage 28, they were fixed with 4% paraformaldehyde (PFA) followed by staining with Phalloidin.

For Centrinone, we incubated stage 14 tethered animal caps in 10 µM Centrinone until they reached stage 25–26 when they were fixed with 4% PFA followed by staining with Phalloidin. We used unmanipulated sibling embryos for staging.

## Immunofluorescence

*Xenopus* embryos were fixed in 2% trichloroacetic acid for 10 min. for anti-Piezo1 antibody labeling.

## Antibody concentrations

Rabbit polyclonal Anti-Piezo1(1:25) antibody from Novus, NBP1-78537, was used to label Piezo1. Alexa 488, Alexa 594, and Alexa 647 (all 1:500) were used as secondary antibodies for immunofluorescence. Alexa 633 and Alexa 488 phalloidin (both 1:50) were used.

## Animal cap dissections

Animal caps were dissected at stage nine as described (*Werner and Mitchell, 2013*). The untethered caps were raised in Danilchik's for *Amy* (DFA) media supplemented with antibiotic/antimycotic in a petri dish, and tethered caps were similarly cultured but, on a slide, treated with fibronectin (25 µg/ml).

## Mechanical stretcher

To subject animal caps to radial tension at a desired force and rate, we developed an animal cap stretcher (Fig. figure supplement 5). Stretcher design, modeling, and initial testing were done in SolidWorks 2017 (Dassault Systèmes). The stretcher is powered by a 1 RPM 12 V DC gear motor geared down to produce an amount of tension that could be used to stretch animal caps in intervals without detachment. The gears converge on a gear strip that pulls an eight-spoked Delrin attachment, which then transduces the motor force to an additional 24 spoked Delrin attachment to produce a tension of 0.48 N in 24 radial directions. Animal caps dissected at St nine were cultured on 0.25 mm thick sheets of silicone (Grace Bio-labs) treated with fibronectin (25 µg/ml) in DFA supplemented with antibiotic/antimycotic. A circle of oil was made around the animal cap and filled with 1/9 MR to keep the animal cap in the culture medium during stretching. The silicone sheet was then affixed to the stretcher via 24 equally radially spaced pins, which connected the sheet to the stretcher. The

stretcher applied 11.5 N of force distributed over the 24 pins in intervals of 1 min on and 10 min off over a total of 180 min per animal cap. All caps that remained attached to the silicone sheet during this process were further analyzed.

## Cell fusion

Embryos were injected with mRNA encoding GFP-SAS6 or Centrin4-RFP and allowed to develop till Stage 21. Chemical fusion was caused by placing the embryos in a solution of 50% polyethylene glycol 4000 (PEG4000) for 20 min, followed by an osmotic shock when the embryos are placed back into 0.1x MMR. After several rinses of fresh 0.1x MMR, the embryos were allowed to recover overnight at 18°C and were fixed in 3%PFA for 1 HR. Embryos were then stained with Phalloidin and DAPI to mark the cell borders and nuclei, respectively. Fused cells were identified as having two nuclei with DAPI signal.

## Image analysis

Images were captured using a Zeiss 710 Live, Zeiss 880, Nikon A1R, or Leica SP8 confocal microscope. Images were processed in Fiji or Adobe Photoshop. Segmentation and 3D reconstruction of intercalating cells and basal bodies were done using IMARIS. For *X. tropicalis* experiments, apical areas were quantified manually, and centriole number were quantified manually or with the analyze particle module in Fiji. For cell fusion, *Cep152* overexpression and *Mcidas* overexpression experiments, centriole numbers, cell size, and nuclei numbers were quantified manually using NIS Elements.

## Quantification and statistical analysis

Statistical significance was performed using GraphPad Prism and is reported in the figures and legends. In all figures, statistical significance was defined at $p < 0.05$. Appropriate sample size (number of embryos and number of cells) was determined based on the previously published data (*Collins et al., 2020*; *Kulkarni et al., 2018a*). Wherever applicable, all experiments were repeated independently two to four times (biological replicates). The comparisons between treatments or species are represented as Violin plots showing all data points and the median. The descriptions of comparisons between treatments are specified in each figure legend. The curve fitting for the regression line was done by statistically comparing the linear model to the second-degree polynomial model to identify the fit that more accurately described the data. Outliers were identified using the ROUT analysis in GraphPad Prism with Q, the maximum desired False Discovery Rate (FDR) = 1%. Outliers were not removed from the data as they did not influence the statistical outcomes of the comparisons. We randomly picked one cell *X. tropicalis* embryos from fertilization as uninjected controls or for MO or RNA injections. Investigators were not blind to experiments or statistical analysis.

## Acknowledgements

We thank Doug DeSimone and Patrick Lusk for discussions and their valuable comments on the paper, Lance Davidson for his valuable input on animal cap and mechanical stretching experiments, and Ellen Su at the Yale Tsai Center for Innovative Thinking for guidance on developing the stretch apparatus used in this work. We also thank the Yale Center for Engineering Innovation and Design for use of instruments in the production of the custom stretch apparatus. We thank the Yale Center for Advanced Light Microscopy for their assistance with confocal imaging. SSK was supported by the NIH Pathway to Independence K99/R00 grant (1K99 HL133606 and 5R00HL133606). MKK was supported by the NIH/NICHD (R01HD102186). JM was supported by the Yale MSTP NIH T32GM007205 Training grant, the Yale Predoctoral Program in cellular and Molecular Biology T32GM007223 Training Grant, and the Paul and Daisy Soros Fellowship for New Americans. RV was supported by a T32 Training grant in Cutaneous Biology (T32AR060710). BJM was supported by NIH/NIGMS (R01GM089970).

## Additional information

### Competing interests

Mustafa K Khokha: is a founder of Victory Genomics. The other authors declare that no competing interests exist.

### Funding

| Funder | Grant reference number | Author |
| --- | --- | --- |
| NIH | 1K99 HL133606 and 5R00HL133606 | Saurabh Kulkarni |
| NICHD | R01HD102186 | Mustafa K Khokha |
| NIH | T32GM007205 | Jonathan Marquez |
| Yale University | T32GM007223 | Jonathan Marquez |
| Paul and Daisy Soros Fellowships for New Americans | | Jonathan Marquez |
| NIH | T32AR060710 | Rosa Ventrella |
| NIGMS | R01GM089970 | Brian J Mitchell |

The funders had no role in study design, data collection and interpretation, or the decision to submit the work for publication.

### Author contributions

Saurabh Kulkarni, Conceptualization, Resources, Data curation, Formal analysis, Funding acquisition, Validation, Investigation, Visualization, Methodology, Writing - original draft, Project administration, Writing - review and editing; Jonathan Marquez, Methodology; Priya Date, Investigation, Methodology; Rosa Ventrella, Investigation; Brian J Mitchell, Investigation, Writing - review and editing; Mustafa K Khokha, Resources, Supervision, Funding acquisition, Investigation, Writing - original draft, Writing - review and editing

### Author ORCIDs

Saurabh Kulkarni (iD) https://orcid.org/0000-0002-0882-6478
Jonathan Marquez (iD) http://orcid.org/0000-0003-3377-7599
Mustafa K Khokha (iD) https://orcid.org/0000-0002-9846-7076

### Ethics

Animal experimentation: *Xenopus tropicalis* were housed and cared for in our aquatics facility according to established protocols approved by the Yale Institutional Animal Care and Use Committee (IACUC, protocol number - 2021-11035) and University of Virginia IACUC (protocol number - 42951119). *Xenopus laevis* were housed and cared for according to established animal care protocol approved by Northwestern University IACUC (protocol number - IS00006468).

### Decision letter and Author response

Decision letter https://doi.org/10.7554/eLife.66076.sa1
Author response https://doi.org/10.7554/eLife.66076.sa2

## Additional files

### Supplementary files

- Transparent reporting form

## Data availability

Data is attached as a source files.

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
