## [Decision Letter]

**Acceptance summary:**

The study of mechanical forces in development is a burgeoning area of cell and developmental biology, and the identification of the roles of pushing and pulling forces in the emergence of frog epidermal multiciliated cells, and the exciting involvement of Piezo1 (and possibly a ciliary population of Piezo1) will be of wide interest. Previous work established a connection between apical area and basal body number, and the implication that Piezo1 is working to adjust those numbers is exciting and raises many interesting and tractable questions.

**Decision letter after peer review:**

Thank you for submitting your article "Mechanical Stretch Scales Centriole Number to Apical Area via Piezo1 in Multiciliated Cells" for consideration by *eLife*. Your article has been reviewed by 3 peer reviewers, one of whom is a member of our Board of Reviewing Editors, and the evaluation has been overseen by Anna Akhmanova as the Senior Editor. The following individual involved in review of your submission has agreed to reveal their identity: Nathalie Jurisch-Yaksi (Reviewer #3).

Essential Revisions:

1) As you will see from the comments below, all of the reviewers requested additional information regarding the application of the statistical tests. There was a lack of clarity as to why various statistical tests were chosen, which statistical tests were employed for which experiment, and how outliers were selected for exclusion.

2) The reviewers sought to know where Piezo1 localizes at the basal body. The presumption is that, as an integral membrane protein, it localizes to the base of ciliary membranes, with implications for how Piezo1 may function in mechanical force sensation. We hope that, given your expertise in superresolution imaging and basal bodies, determining whether Piezo1 co-localizes with the ciliary membrane is feasible using advanced light microscopy.

3) A central conclusion of your work is that mechanical stretch is acting through Piezo1 to regulate centriole number in multiciliated cells. The reviewers request that to directly show that mechanical forces are acting through Piezo1 remove/reduce Piezo1 function in the context of the stretched and tethered animal caps. A strong prediction of your work is that if mechanical forces act through Piezo1 to increase centriole number, then reducing Piezo1 (by morpholino, CRISPR KO or GSMTx-4) followed by stretching of the animal caps should attenuate the increase in centriole number. This experiment will more directly test the hypothesis than embryo KO/KD experiments, as you aptly note in your manuscript.

*Reviewer #1:*

In their manuscript, "Mechanical Stretch Scales Centriole Number to Apical Area via Piezo1 in Multiciliated Cells", Kulkarni et al., examine the relationship between mechanical stretch, apical area and centriole number in differentiating multiciliated cells of the frog epidermis. They demonstrate that there is a positive correlation between centriole number and apical area of multiciliated cells and that this relationship holds true across several perturbations of cell size and across two species of *Xenopus* of different sizes. The authors also show that the mechanical stress on multiciliated cells affects centriole number and that this mechanical stress may be sensed by a mechanosensitive channel, Piezo1, which localizes to centrioles. This work builds on prior work by Nanjundappa et al., 2019 to suggest that Piezo1 is a determinant sensing mechanical forces which correlate centriole number with apical surface area.

Overall, the conclusions of the manuscript are well-supported by the presented data. There are a few specific issues that should be addressed prior to publication:

1) One of the major novel conclusions that the authors reach is that mechanical stretch is acting through Piezo1 to regulate centriole number in multiciliated cells. While the claim that the mechanical tension regulates the apical area of the cell and the number of centrioles is well-supported, it is unclear whether the mechanical tension is sensed by Piezo1 in this system. As the authors note, genetic perturbations, especially in a developing embryo, can have effects that are indirect. To show that mechanical forces are acting through Piezo1, the authors should remove/reduce Piezo1 function in the context of the stretched and tethered animal caps. If mechanical forces on the cell act through Piezo1 to increase centriole number, then reducing Piezo1 (by morpholino, CRISPR KO or GSMTx-4) followed by stretching of the animal caps should no longer increase centriole number.

2) The author's conclusion that "Piezo1 is essential for centriole amplification," is an overstatement. Piezo1 inhibited cells still amplify centrioles, just not to the same extent as the control cells.

3) The authors increase centriole number and observe a concomitant expansion of apical area, and decrease centriole number and observe no concomitant decrease in apical area. From these observations, the authors conclude that apical area does not depend on centriole number. This conclusion would seem to disregard the results with increased centriole number. Perhaps a more precise hypothesis resulting from these observations would be that some minimal apical area is independent of centriole number, and that centriole amplification may be sufficient to expand the apical domain (at least in the presence of Piezo1).

Where is Piezo1 at the basal body? The localization to centrioles and its identity as an integral membrane protein suggests that it is at the base of ciliary membranes? The authors have conducted high resolution imaging of similar proteins at the base of cilia in the past, and elucidating the precise localization of Piezo1 will help explain how Piezo1 may be regulating centriole number. For example, examining Piezo1 localization in multiciliated cells during centriole synthesis and migration would be expected to demonstrate Piezo1 localization to centrioles after the centrioles dock to membranes. A time series of Piezo1 localization could also support the assertion that Piezo1 regulates a late stage of centriole synthesis, as suggested in the model in Figure 4i.

Also, for the statistical tests, it is not clear when unpaired t-tests, non-parametric Mann-Whitney tests, parametric Brown-Forsythe and Welch ANOVA tests, and non-parametric Kruskal-Wallis tests are used. Perhaps the tests used can be indicated in the figure legends. Can the statistical tests be applied without removing outliers to increase confidence in the results?

The purple line bisecting the MCC progenitor in Figure 1A could be read as a cell membrane or division plane, although I think the intent was to indicate the third dimension. Perhaps shading could be used instead?

In Figure 1i, the authors combine data from laevis and tropicalis to illustrate a correlation between centriole number and MCC apical area. Is this correlation not significant for data from a single organism? Combining the organisms assumes that the biological principles dictating apical area and centriole number are the same in both (a reasonable assumption given the evolutionary closeness between laevis and tropicalis).

I recognize that the authors have a preprint in support of a role of ccdc11 in cytokinesis (S. Kulkarni et al., 2018). However, as ccdc11 is a ciliary and centriolar satellite component involved in left-right axis patterning, knockdown of ccdc11 may affect more than just cytokinesis. Is it possible the authors generate large cells by inhibiting a more dedicated regulator of cytokinesis?

The decrease in TR to 0.8 as the apical area increases to ~300 μm2 is suggested to indicate a shift from cell autonomous pushing forces to cell non-autonomous pulling forces. It would also be consistent with a shift in the stiffness of surrounding cells to from uniform to nonuniform, no?

Can the authors calculate R2 values for the data included in Figure 1 supplement 1?

Does the imaging regarding centriole biogenesis during intercalation in Figure 2 suggest that there are two waves of centriole synthesis – one wave prior to intercalation and a second after the cell has integrated into the apical epithelium – or is it a continuous process?

The phalloidin staining of untethered and tethered animal caps shown in Figure 3 supplement 2 are markedly different. Is the fibronectin provided in the tethered animal caps also preventing apoptosis, as has been noted previously for other cell types?

In non-MCCs, Piezo1 is expressed at cell junctions, depicted in Figure 4 supplement 1b. However, this cell junction localization is not apparent in MCCs. Is Piezo1 expressed at cell junctions of MCCs?

Was a nonspecific CRISPR guide used to control for the Piezo1 sgRNA result?

Can the authors rescue centriole number in untethered animal caps by a Piezo1 agonist such as Yoda1 (Syeda et al., *eLife*, 2015)?

Does the chemical method of fusing neighboring cells fuse two cells or more cells?

*Reviewer #2:*

This study utilizes multiple lines of evidence to establish that the mechanical stretching across an epithelium concurrent with apical expansion of multiciliated cells (MCCs) helps to regulate the expansion of the centriole/basal body, which are key organelles necessary for the apical extension of cilia from these cells. This is an important process to understand as disruption of ciliary axonemes from the apical surface can result in common disease pathologies collectively calleds ciliopathies in particular for those affecting the multiciliated tissues lining the airway, reproductive tracts, and cerebrospinal fluid flows in the brain and spinal cord. The work started by making several correlations between apical size with centriole numbers at the apical surface both due to natural cell size variation comparing two *Xenopus* species. They go on test if the relationship of centriole number and apical area would persist if apical area was increased using a variety of complementary approaches to show this relationship remained robust. They next tested whether stimulation of centriole numbers would drive apical expansion, showing that overexpression of cep152 does, as expected, increase centriole numbers, yet also greatly increased the apical area of the MCCs. Conversely when they reduced centriole expansion using a drug, Centrinone, the apical area of MCCs was largely unaffected. Altogether, suggested that regulation of apical area is not dependent on centriole numbers.

Next, they set out to test if apical area helps to 'fine-tune' the numbers of centrioles. They set up a mechanical stretching approach using animal cap explants showing that stretching the animal caps, which resulted in increased apical area and increased centriole numbers compared to unstretched animal caps, suggesting a mechanotransduction may drive this relationship. They then showed that a mechanosensitive ion channel protein Piezo 1 is present at cell junctions and localized adjacent to centrioles at the apical surface of MCCs and demonstrate that Piezo1 coordinates the expansion of centriole numbers at the apical surface of MCCs in response to apical expansion of these cells and stretching of the epithelium. The conclusions of this paper are mostly well supported by data, but some aspects of data analysis and reporting can be strengthened.

In general, the authors do not clearly explain the majority of reported values or address statistic in the manuscript. The beginning of the study reports a median value but as we continue onwards there is no consistent indication of whether the numbers reported are the median or mean, finally by the end (page 13) there are just numbers in a paratheses. This is very inconsistent and gives the appearance of indiscriminate methodology. I recommend that all numbers reported makes a clear indication of what the value is, with error. In addition, I found it difficult to find the statistical tests used each analysis, please report this in the figure legends at minimum.

Centriole expansion after overexpression of cep152 is strongly correlated with and with increased apical expansion. In contrast, loss of function of piezo1 only mildly affects apically docked basal body/centriole numbers without affecting apical area in some approaches, while increasing apical area it in other approaches. Why is this? Despite the results showing reduced centrioles (after Centrinone treatment) did not affect apical area, I wonder if aberrantly increasing centriole numbers does have a role, via Piezo 1, for driving apical expansion? I think a simple test of this would be to combine the overexpression of cep152 along with blockade of Piezo 1 function (both chemical and genetic) to tease apart this issue. If Piezo 1 is important for coordination of centriole expansion during epithelial stretching then it should limit the function of cep152 overexpression in this approach. Alternatively, If blockade of Piezo 1 blocks apical expansion after cep152 overexpression, then this further supports your model to my mind.

Need to clean up the genetic nomenclature. If you are expression CEP152 from synthetic RNA, maybe just state that (as Cep152) instead of implying that you are overexpressing the CEP152 as a protein. Moreover, this use of CEP152 is not consistent with your nomenclature of Piezo 1. Recommend cleaning up your nomenclature based on the *Xenopus* guidelines: (https://www.xenbase.org/gene/static/geneNomenclature.jsp0). For *Xenopus* the RNA symbols are the same as gene symbols in lowercase and italics and match human symbol nomenclature.

*Reviewer #3:*

In this manuscript, the authors address how multiciliated cells count the numbers of cilia/centrioles. The authors discovered that the number of cilia/ centrioles depend on the cell surface and the mechanical forces exposed on the cells, by using a large series of manipulations (genetic and physical) to support their claims. The authors show that the mechanosensor Piezo1 is involved in this process as inhibition or genetic knock-down of Piezo1 abrogate the relationship between size of the cell and number of centrioles. Altogether, this manuscript goes beyond the current literature by providing a mechanistic understanding of a fundamental cell biology question.

Strength:

The authors are utilising their model to a full extent. The manipulations done by the authors addressed the problem from many angles, which strengthen their results.

Weakness:

Some of the analyses performed by the authors need further clarifications. It is important to note that this does not invalidate the authors' claims since the effects described are very strong and obvious.

I am supportive of the publication of this work, but I have few comments as indicated below that need some particular attention

1. The authors perform a series of regression analysis to identify potential correlation between size and number of centrioles. I have few comments regarding these analyses:

– As indicated in their material and method section, the authors decided to perform either a linear regression or a second degree polynomial model depending on which model fitted better the data. I fear that using different types of regressions throughout the manuscript have strong consequences on the interpretation of the data. Figure 1C is a very good example in my opinion. I can observe that the slope of the curve for the immature cells (in green) is high, while the slope for mature cells (in blue) is much smaller. I suggest the authors to consider these two categories of cells differently and fit a curve for the immature cells and for the mature cells.

– I understand that most of the quantifications were done in mature cells and not in immature cells, upon the various manipulations. Yet the authors overlay the distribution of immature/mature cells shown in Figure 1C on those graphs. I suggest the authors to compare mature cells only with mature cells and drop the immature X tropicalis cells from the various plots throughout the manuscript.

– It is in few instance not clear what datapoints were used for the regression analysis. I suggest the authors to clarify this in the figure legend of using clearly defined colorcode.

2. The authors describe piezo to be the sensor transforming the mechanical stretch into the amplification of centrioles. If this were true, I would suppose that there is no difference in centriole numbers in intercalating/immature cells upon piezo inhibition, but that difference only appear after the cell is intercalated and stretched. Could the authors clarify whether (1) they measured number of centrioles only in mature cells or not? (2) They report between 100-120 centrioles upon piezo manipulation. Is this number similar to the control immature cells? (3) Is piezo already present in the vicinity of the centriole in immature cells or does its localization correlate to the maturity of the cell?

3. The authors do not provide any further insights on how piezo does regulate centriole amplification, and I do understand that these experiments are out of scope of this manuscript. It would be nice if the authors could at least include in their discussion some potential suggestions/references on how they expect this to happen

---

## [Author Response]

Essential Revisions:1) As you will see from the comments below, all of the reviewers requested additional information regarding the application of the statistical tests. There was a lack of clarity as to why various statistical tests were chosen, which statistical tests were employed for which experiment, and how outliers were selected for exclusion.

We have now provided a better explanation for which statistical tests were chosen in each figure legend. Of note, we have not removed any outliers from any of the analyses. While our experimental disruptions can be significant with occasional outliers, we did not exclude these data points. The results are sufficiently robust that we can still see clear trends supported by our statistical testing.

2) The reviewers sought to know where Piezo1 localizes at the basal body. The presumption is that, as an integral membrane protein, it localizes to the base of ciliary membranes, with implications for how Piezo1 may function in mechanical force sensation. We hope that, given your expertise in superresolution imaging and basal bodies, determining whether Piezo1 co-localizes with the ciliary membrane is feasible using advanced light microscopy.

Now that we have established that Piezo^^1^ localizes to the cilium, its precise localization is a very interesting question. This image of Piezo1 labeling in *Xenopus* MCCs from animal caps was generated using a NIKON SoRa followed by deconvolution. Achieving this super-resolution image of Piezo1 has proven to be very challenging for several reasons. In our previous work (Del Viso *et al.* Dev Cell 2016), we exploited monolayers of cultured cells which have optimal, optical clarity. Here, while *Xenopus* animal caps are exceptional for mechanical stretching, gene product manipulation, and rapid growth of MCCs, the animal cap is composed of a few layers of cells. This is not a problem for standard confocal microscopy, but once we pushed towards super-resolution microscopy, techniques like PALM/STORM become much more challenging. We also attempted NIKON STED microscopy to achieve higher resolution (50-60 nm) than SoRa (120 nm). However, pigmentation in the *Xenopus* epithelial cells absorbs heat from the intense laser exposure and burns the cells preventing us from collecting the STED data.

Nevertheless, with SoRa, we can see that Piezo1 makes a ring of roughly 250 nm. Of course, this result leads to multiple interesting questions: 1) where in the context of the basal body does this ring lie? Does it lie next to the basal body (unexpected) or does it surround the basal body? 2) Is Piezo1 present in the peri-ciliary membrane or present in the ciliary membrane, at the level of transition zone? 3) How does Piezo1 traffic to the base of cilia? These are all critical questions that have important implications for the formation of the Piezo1 ring and Piezo1 function during ciliogenesis.

To address some of these questions, we need to co-localize Piezo1 with other basal body/cilium related landmarks. While we are attempting to co-localize some of the known basal body markers like Centrin and Chibby, we have not had any success so far because of a few challenges. First, to visualize Piezo1, we used TCA fixation (after testing numerous permeability/ fixation methods to see which is optimal). However, we found that TCA depletes GFP/RFP signal which we commonly use to visualize these basal body proteins (works well with PFA fixation). We had hoped that we could quickly provide the reviewers an answer on the localization of Piezo1. However, at this point, we will need to transition to using cilia antibodies (most of which have not been tested with TCA fixation). Additionally, we may need to move to mammalian MCC culture (more of a monolayer where a host of antibodies could be exploited) for super-resolution microscopy as well as immuno-EM (which has another host of challenges). Saurabh Kulkarni (the first author) has recently moved to UVa where his lab is continuing to pursue the localization and role of Piezo1 in MCCs. He is also working on domain analysis experiments to identify regions of Piezo1 that direct its localization to cilia as well as the dynamics of Piezo1 localization (Does Piezo1 localize to a newly duplicated basal body in the cytoplasm or after basal body docking?). While we appreciate the reviewers’ curiosity about Piezo1 localization (as we do too!), we feel that the combination of these results with super-resolution imaging of Piezo1 would be better for a subsequent manuscript where we can combine all these experiments into a comprehensive manuscript. We do believe that Piezo1 localization to the cilium base really opens a whole new avenue of research on the mechanochemical sensation of cilia.

3) A central conclusion of your work is that mechanical stretch is acting through Piezo1 to regulate centriole number in multiciliated cells. The reviewers request that to directly show that mechanical forces are acting through Piezo1 remove/reduce Piezo1 function in the context of the stretched and tethered animal caps. A strong prediction of your work is that if mechanical forces act through Piezo1 to increase centriole number, then reducing Piezo1 (by morpholino, CRISPR KO or GSMTx-4) followed by stretching of the animal caps should attenuate the increase in centriole number. This experiment will more directly test the hypothesis than embryo KO/KD experiments, as you aptly note in your manuscript.

We attempted to answer this question using two different experimental setups. First, we compared the animal caps that are untethered (free-floating) and tethered to the slide using fibronectin. Tethered cells exert outward pulling forces stretching the epithelia which are reduced in the free-floating caps. We compared the apical area and centriole number in these two conditions in response to Piezo1 KD. In the untethered animal caps, depletion of Piezo1 did not affect the generation of ~100 centrioles (no significant difference from control untethered animal caps). In contrast, in the tethered animal caps, Piezo1 KD limited centriole amplification compared to control tethered animal caps. In addition, we discovered an interesting role of Piezo1 in regulating the apical area. The apical area in untethered animal caps did not significantly differ between controls and Piezo1 KD treatment. On the contrary, in tethered animal caps, Piezo1 depletion led to a significant *increase* in the apical area compared to controls. Therefore, in the context of stretch, Piezo1 plays a role in regulating apical area as well as centriole amplification. Effectively, depletion of Piezo1 uncouples apical area and centriole amplification when the tissue undergoes outward pulling forces.

Next, we attempted to mechanically stretch Piezo1 depleted animal caps to test its role in stretch-induced centriole amplification. We attempted the experiment a few times; however, each time it failed because the Piezo1 depleted animal caps dissociated during stretching. It appears that Piezo1 is critical for maintaining junctional integrity in epithelial tissue and loss of Piezo1 leads to weakening of cell contacts. Consequently, the animal cap tears preventing us from completing the experiment. To overcome this challenge, we need to stretch the cells more slowly and in finer increments which unfortunately is not possible with our DIY stretcher. Of course, in the context of tethered animal caps with Piezo1 depletion, the cells can withstand stretching suggesting that a more gradual mechanical stretch should work. Saurabh has purchased a more sophisticated stretcher in his own lab which might be able to accomplish this task. However, the new stretcher is under production and will take at least a few months to arrive.

Nevertheless, we have added the data on Piezo1 depletion in tethered and untethered animal caps to the manuscript (NEW Figure 6) to address the question of Piezo1 depletion and centriole amplification in the context of stretch.